# Glucose and Lipid Profiles Predict Anthropometric Changes in Drug-Naïve Adolescents Starting Treatment with Risperidone or Sertraline: A Pilot Study

**DOI:** 10.3390/biomedicines11010048

**Published:** 2022-12-25

**Authors:** Emilia Matera, Gloria Cristofano, Flora Furente, Lucia Marzulli, Martina Tarantini, Lucia Margari, Francesco Maria Piarulli, Andrea De Giacomo, Maria Giuseppina Petruzzelli

**Affiliations:** 1DiMePRe-J—Department of Precision and Rigenerative Medicine—Jonic Area, University of Bari “Aldo Moro”, 70100 Bari, Italy; 2DiBraiN—Department of Translational Biomedicine Neurosciences, University of Bari “Aldo Moro”, 70100 Bari, Italy

**Keywords:** second-generation antipsychotics, risperidone, sertraline, selective serotonin reuptake inhibitors, metabolic risk, anthropometric changes, glucose profile, lipid profile, weight gain, drug-naïve adolescents

## Abstract

Psychiatric disorders are associated with cardiometabolic diseases, partly due to adverse drug effects with individual risk variabilities. Risperidone and sertraline are widely used for youths. Although they may be exposed to anthropometric changes, few data about this population exist. We evaluated the correlation between several blood parameters and body changes in a very small group of drug-naïve adolescents who had started risperidone or sertraline. We examined weight, waist circumference (WC), WC/height ratio and body mass index (BMI) at baseline (T0) and after at least three months of therapy (T1), and blood glucose and lipid profiles at T0. Here, we show significant increases in several anthropometric parameters in both groups, a negative correlation between HDL and ΔWC in the risperidone group and positive correlations between insulin and ΔBMI and between HOMA-IR and ΔBMI in the sertraline group. Despite the sample size, these results are important because it is difficult to study adolescents who are long-term-compliant with psychotropic drugs. This pilot study supports the importance of future large-scale investigations to understand the metabolic risk profiles of psychotropic drugs, their individual vulnerabilities and their underlying mechanisms. Simultaneous guideline-based psychiatric and metabolic interventions should be part of daily practice.

## 1. Introduction

In recent years, the use of psychotropic drugs in children and adolescents for treatment of many psychiatric conditions has grown worldwide, especially that of selective serotonin reuptake inhibitors (SSRIs) and second-generation antipsychotics (SGAs), often with long-term intake [1]. Despite their undoubted effectiveness, prescriptions of both SSRIs and SGAs often exceed the authorized indications under the age of 18, resulting in wide off-label use as a necessary part of treatment to optimize clinical outcome and increase chance of recovery [2]. The increase in the number of prescriptions and the frequent off-label use imply careful assessment of the risk/benefit ratio before prescription of SSRIs and SGAs, as well as regular monitoring of adverse effects to optimize patient care. Indeed, use of these drugs for children and adolescents not only sets up an already vulnerable population for side effects at an early age but also represents a significant gap in the literature, especially due to the scarcity of safety- and tolerability-controlled clinical studies with respect to adulthood [3,4].

Antipsychotic-induced weight gain has been recognized as a major health concern in children and adolescents, and the emerging literature has shown that metabolic adverse effects related to weight gain exacerbate the inherent risks of metabolic syndrome, type 2 diabetes onset and cardiovascular disease in the long term [5]. Multiple mechanisms have been hypothesized to influence pediatric antipsychotic-induced weight gain and the metabolic effects of SGAs. It is likely that adverse events are due to a combination of different mechanisms, including SGA influence on receptor signaling and hormone mediation, predisposition due to genetic risk factors and SGA effects on the gut microbiome [6]. Besides the receptor binding profile, the observed weight gain in youths starting antipsychotic treatment was highly heterogeneous, with some youths gaining a lot of weight, while others gain little. Most of the weight gain was reported to occur in the first weeks of treatment, and the patients most at risk were younger, had lower body mass indexes (BMIs) and were female [7,8]. This heterogeneity suggests that certain patient-related factors underlie the risk for antipsychotic-induced weight gain. Identification of these risk factors is an important target in obesity prevention, as it can facilitate early recognition and interventions for children and adolescents at risk [9].

Likewise, we know that weight gain is also a frequent side effect of SSRIs, but to date, we have less clarity about the relationship between prolonged use of SSRIs and the risk of cardiometabolic diseases such as metabolic syndrome, type 2 diabetes and cardiovascular disease, also because the mechanisms of weight gain related to antidepressant use are not well known. Concerns exist that SSRI use increases the risk of developing type 2 diabetes in adults, but evidence of this in children and adolescents is limited. SSRIs are generally associated with modest weight gain, but this may be sufficient in transitioning some patients from normal weight to overweight/obesity and elevate the risk of type 2 diabetes [10]. Furthermore, it is possible that behaviors associated with depression severity may contribute to the observed risk of weight gain. Data regarding this potential safety concern in children and adolescents is limited so that in the absence of randomized clinical trials, evidence must be generated using real-world data, as it must to examine specific SSRIs.

We know that each drug has its own specific risk profile, but high interindividual variability in weight gain among patients treated with a given agent suggests that other subjective risk factors play a role in adjusting individual propensity to weight gain. Therefore, finding out biomarkers of early identification of high risk of weight gain would be very useful in clinical practice to optimize tolerability and adherence to treatment, especially in young patients. Although a lot of studies have shown impairment of glucose tolerance and unfavorable lipid change after treatment with psychotropic drugs, to our knowledge, no studies have explored the role of baseline metabolic status as a specific risk factor that favors weight gain during the early stages of treatment with SGAs and SSRIs in adolescents [11]. Considering that metabolic adverse effects differ by class and for individual drugs within the same class, the present study started from the hypothesis that biochemical baseline parameters of glucose and lipid metabolism could be associated with weight-gain heterogeneity in adolescent patients who were starting first-time treatment with SGAs and SSRIs. Thus, we choose to study the relationship between the biochemical baseline parameters of glucose and lipid metabolism and modification of anthropometric parameters associated with risperidone and sertraline treatment, considering that these are the most widely used SGAs and SSRIs, respectively, in the pediatric age group. Risperidone is one of the most widely used SGAs in children and the adolescent population, both on- and off-label, and according to the US FDA Adverse Event Reporting System (FAERS), between 2012 and 2021, it was frequently associated with weight gain in adult patients [12]. As shown by previous studies published by our group and by other authors [13,14], risperidone induced significant increases in weight and also variations in anthropometric and metabolic parameters in children and adolescent patients after a few months of therapy. Sertraline is a widely used SSRI for patients under the age of 18, on- and off-label, for the reasons of its safety and tolerability profiles and its good efficacy as an antidepressant and an antianxiety agent [15,16], but data on the relationship between sertraline treatment and childhood weight gain are not known. For the purpose of this study, we assessed a set of blood biochemical parameters related to glucose and lipid metabolism in two different clinical groups of adolescent candidates for treatment with risperidone or sertraline according to clinical judgment for symptomatology and diagnosis. As the first aim of study, we evaluated separately, in the two study groups, modifications of anthropometric parameters (weight, waist circumference, waist to height ratio, body mass index) after treatment with risperidone or sertraline for a period ranging from 3 to 6 months in order to search for significant difference between T0-T1. The second aim was to study the correlation between the baseline levels of the blood biochemical parameters of glucose metabolism (fasting glucose, fasting insulin, HOMA IR) and lipid metabolism (total cholesterol, triglycerides, HDL, LDL) with the T1-T0 differences of the anthropometric parameters (Δ Weight, Δ Waist circumference, Δ Waist to height ratio, Δ BMI).

## 2. Materials and Methods

### 2.1. Subjects

A prospective chart review was performed of inpatients of both sexes from the Child and Adolescent Neuropsychiatry Unit of University of Bari “Aldo Moro”, Italy, who have never taken psychotropic drugs and who started drug therapy with risperidone or sertraline in the period between October 2021 and October 2022. Moreover, the recruitment process of our sample included a three- to six-month follow-up evaluation after enrollment. All patients met the following inclusion criteria: (a) age of up to 18 years, (b) a neuropsychiatric diagnosis that requires risperidone or sertraline, (c) drug naïveté for psychotropic drugs, and (d) receipt of risperidone or sertraline continuously and with adequate compliance for at least 3 months. Patients who had lifelong psychotropic drug exposure, present or lifelong pathology that could affect body weight (e.g., eating disorders, Prader–Willi syndrome), cardiovascular abnormalities, or clinical or biochemical evidence of acute medical disorders were excluded from this study. The neuropsychiatric diagnoses were formulated by a child neuropsychiatrist according to the criteria of the Diagnostic and Statistical Manual of Mental Disorders, Fifth Edition (DSM-5) [17] through clinical interviews with patients and their families, medical history and clinical observation.

### 2.2. Assessment and Measurements

All patients underwent a baseline evaluation (T0), which included a collection of data regarding family and personal history of metabolic disease (obesity, glucose intolerance and diabetes, dyslipidemia, cardiovascular and thyroid disease) and other past medical problems, physical and neurological examinations, anthropometric parameters and blood tests, in addition to recommended therapy to be undertaken and its dosage. Moreover, the same patients underwent a follow-up (T1) assessment after at least three months of continuous therapy with risperidone or sertraline. The follow-up assessment included revaluation of anthropometric and blood-test data in addition to possible dosage changes for drug therapy. During the drug treatment, clinical control experiments were performed with a frequency variable in order to make any dosage corrections with respect to the clinical symptoms of the patients.

#### 2.2.1. Anthropometric Parameters

The following measurements for anthropometric data were taken during the baseline and follow-up times: weight (in kilograms), height (in centimeters), waist circumference (WC, in centimeters), waist-to-height ratio (WHtR) and BMI (kg/m^2^). Height was measured without shoes, using a stadiometer Marsden 250P (Charder Electronic Co. Ltd., Taichung, Taiwan). Weight was measured without shoes, jackets or heavy sweaters and with empty pockets, using a GIMA Astra 27 310 scale that had been calibrated every 3 months by a trained team. WC was measured via a tape measure placed at the midpoint between the last rib and the iliac crest of each patient at the end of a normal exhalation. WtHRs were established using a calculator available online (https://www.mytecbits.com/tools/medical/waist-height-ratio-calculator, accessed on 1 October 2021) [18]. These anthropometric parameters were chosen among the core elements of anthropometry and measured according to international recommendations from the American Academy of Pediatrics [19].

#### 2.2.2. Blood Biochemical Parameters

Blood samples were collected after patients had fasted overnight and before they took the drugs, according to a standardized protocol for venous-blood-sample collection used in our clinic, in order to safeguard testing reliability [20].

Serum glucose was determined using an enzymatic method. Serum insulin was estimated with chemiluminescence. HOMA-IR was calculated using the HOMA-IR calculator, version 2.2.3, provided by Oxford University (a free download is available from the website www.dtu.ox.ac.uk) [21]. Total cholesterol levels were measured through a standardized method that is traceable to the International Federation of Clinical Chemistry Working Group (IFCC-WG) Reference Method. HDL was estimated with a clearance assay. LDL was determined using the Fridewald formula [22]. Glucose, insulin and lipids were analyzed at the Clinical Pathology laboratory of the University Hospital of Bari.

### 2.3. Statistical Analyses

All of the variables were recorded in a structured form specifically for this research. An R statistical environment, version 4.1.2 (The R Foundation for Statistical Computing; Vienna, Austria), was used to analyze data [23]. Descriptive analyses were produced for sociodemographic and clinical characteristics and for anthropometric and biochemical parameters, including frequencies, means and standard deviations. According to the verification of normality assumption through the Shapiro–Wilk test, comparisons between unpaired samples were made with independent Student’s *t*-tests or nonparametric Mann–Whitney tests, and T0-T1 comparisons were made through paired Student’s *t*-tests or nonparametric Wilcoxon tests for paired samples. Moreover, to study the correlations between the variations of anthropometric parameters and baseline biochemical parameters, Pearson’s or Spearman’s coefficients, according to their distribution, were examined for both groups. Statistical significance was set for *p*-values < 0.05.

## 3. Results

### 3.1. Sociodemographic and Clinical Features of the Samples

This study sample included 22 patients in the risperidone group (twelve males and ten females, aged 12.7 ± 2.8 years) and 18 patients in the sertraline group (four males and fourteen females, aged 14.4 ± 1.5 years). Their risk factors for metabolic diseases, main diagnoses, mean dosages and treatment durations are summarized in Table 1.

### 3.2. Evaluation of Baseline Anthropometric and Haematochemical Parameters: Comparison between the Risperidone and Sertraline Groups

When we statistically compared the baseline anthropometric and blood biochemical parameters between the risperidone and sertraline groups, we did not find significant differences (see Table 2).

### 3.3. Evaluation of Changes in Anthropometric Parameters after Treatment with Risperidone and with Sertraline

The paired comparisons between T0 and T1 in the risperidone and sertraline groups showed significant differences in WC, WHtR and BMI in both groups (see Table 3).

### 3.4. Correlation between Baseline Haematochemical Parameters and Variation of Anthropometric Parameters in the Risperidone Group

In a correlation analysis, baseline HDL concentration was negatively correlated with variation in WC in this group. No other significant correlation was found (see Table 4).

### 3.5. Correlation between Baseline Haematochemical Parameters and Variation of Anthropometric Parameters in the Sertraline Group

In correlation analysis, baseline insulin concentration and HOMA-IR were positively correlated with BMI variation in this group. No other significant correlation was found (see Table 5).

## 4. Discussion

In the present study, we evaluated a set of blood biochemical parameters in two drug-naïve groups of adolescents who had started risperidone or sertraline therapy, with the aim to investigate whether the baseline biochemical parameters of glucose and lipid metabolism could be associated with an increased risk of weight gain.

The first result we obtained was a significant increase in the mean values of the anthropometric parameters of weight, BMI, WC and WtHR between T0 and T1 in both the risperidone and sertraline groups. Patients with severe mental illness are known to have higher morbidity and mortality than those of the general population, often secondary to onset of metabolic disorders caused by treatment with psychotropic drugs, and SGAs are the most studied group among these drugs [24,25]. Increase in weight and the associated anthropometric changes are frequent and early adverse effects, especially in children and adolescents, probably due to less prior SGA exposure [13,14,26]. The mechanisms by which SGAs induce these events have been related to modulation of adrenergic, muscarinic and cholinergic receptors of serotonin, dopamine and histamine in the central nervous system. SGAs act on adiponectin, ghrelin, insulin and leptin, which affect appetite and nutrition, energy expenditure and metabolic rate. In addition, SGAs have been associated with perturbations of intestinal microflora, but the link between these changes and weight gain is only beginning to be explored [6,27]. Even though the results of several studies have suggested that SSRIs have neutral effects on weight, other authors have shown that antidepressants such as SSRIs may also be associated with increased incidence of weight gain, abdominal obesity and BMI increase, but with inconsistent explanations for this association [15,24,26,28,29,30,31,32]. Metabolic abnormalities that are secondary to weight gain and obesity, induced with the use of both SGAs and SSRIs, included dyslipidemia, insulin resistance, diabetes and metabolic syndrome [6,24,33,34], which cause considerable concern because of their increased risk for cardiovascular and respiratory diseases (stroke, heart attack, sleep apnea), joint problems (pain caused by osteoarthritis of the knees, hips and back), gynecological disorders (menstrual irregularities, polycystic ovary syndrome), digestive system diseases (gastroesophageal reflux, gallbladder stones) and certain types of cancers (endometrial, colorectal, gallbladder and breast cancer). We must consider that the risk of these pathologies is higher in early-onset cases and in persistence of the predisposing factors over time; moreover, treatment with psychotropic drugs is generally long-lasting, and younger patients are more sensitive to weight gain induced by psychotropic drugs, but until now, this had not been not supported by any underlying molecular mechanism [35].

The other results we found were that in the risperidone group, the baseline serum HDL level was negatively related to changes in WC, while in the sertraline group, baseline serum-insulin and HOMA-IR levels were positively related to changes in BMI after drug therapy. More specifically, in the risperidone group, we found that baseline HDL levels, which are known to have anti-inflammatory action [36], were a protective factor against metabolic risk. Young people with lower LDLs, which are indicative of healthier lifestyle habits, would have a greater tendency to gain weight induced by antipsychotics, while young people with an already poor diet, a low level of physical activity and a low resting metabolic rate at baseline could suffer less of this effect, as there may be a limit to how low resting metabolism and how poor diets and physical-activity levels can be [35]. In the sertraline group, the baseline levels of HOMA-IR and insulin were factors that increased metabolic risk. These data support the idea that individual and environmental factors may underlie the anthropometric variations associated with SGA and SSRI treatment . Several pre-existing data showed significant variability in propensity of antipsychotic-and antidepressant-induced weight gain and other metabolic alterations, some of which may be genetically mediated [37]. Several studies show that, during early stages of illness and before initiation of treatment, patients with first-episode psychosis have high risk of weight gain, glucose dysregulation and dyslipidemia, which further increases cardiovascular risk after exposure to antipsychotic drugs [38,39,40,41]. Weight gain could be associated with a specific class of triglycerides produced de novo by the liver, but not the dietary lipids that may also contribute to development of metabolic comorbidities potentially independently of antipsychotic medication [40,42]. Recent attention has also been focused on metabolic dysregulations including abdominal obesity, lipid dysregulation and hyperglycemia, which could be associated with onset of depression, bipolar and anxiety disorders and their chronicity [43,44]. For depression disorder in particular, these relationships were stronger among females than males [45], while conflicting results emerged for psychosis [46]. The possibility of a common pathophysiology that underlies weight gain and its associated metabolic anomalies and mental illness, including schizophrenia and depression, was also examined; several genes that are shared between mental disorders and obesity have been identified, including FTO, POMC, ITIH4, TLR4, BDNF and CREB1. However, those results have so far been inconclusive, mainly due to polygenic contribution to these diseases, which makes it difficult to identify allelic risk variants. About the bidirectional association between depression and diabetes, it has been suggested that impaired insulin signaling, including insulin availability, is important to the pathophysiology of depression; this was supposed according to the antidepressant properties of drugs such as insulin and metformin [16,44,47]. Other proposed biological mechanisms that could underlie mental disorders and obesity include dysregulated hypothalamus–pituitary–adrenal axis function, inflammatory processes and hormonal and neurotransmitter imbalances that generally overlap with neurohormonal changes that are typical of the pubertal period [48]. Lastly, the fact that weight and general anthropometric changes in psychotic and depressed nonmedicated individuals could be influenced by unhealthy lifestyles and eating patterns (high total energy and energy-dense and processed food intake), sedentary activity, prolonged total time with screens, smoking and low income, should not be underestimated [48,49,50]; this unfortunately was not evaluated in our analysis. It is also possible that anthropometric variations in children and adolescents treated with psychotropic drugs may depend on follow-up duration. Weight gain has been reported to be the most pronounced during the first few weeks of SGA use, stabilizing during continued treatment, although long-term data are limited [9].

The first limitation of this study was the presence of diagnosis nonhomogeneity that could have influenced our results, considering that the available literature concerns mostly patients with psychosis and depressive disorders [9,38,39,40,41,45,47,48,49,50]. In addition, the lack of lifestyle information, the very small sample size, the short period of drug treatment and follow-up, the absence of a healthy control group and the fact that this study was conducted during a time when there were still substantial changes in household lifestyles due to the COVID-19 pandemic prevented us from drawing definite conclusions about the obtained results.

Despite these numerous limitations, this study confirms the pre-existing data of the onset of anthropometric and metabolic alterations secondary to psychotropic drug treatment in the developmental age. These data are still limited in the juvenile population as compared to the adult population, but are more numerous than in the past. For example, it is known that younger patients, patients with lower BMI and female patients who use antipsychotics are at greater risk of weight gain or metabolic changes [7,8,26,51]. The literature data of individual factors that could predispose anthropometric variations at risk of evolution into metabolic pathologies before treatment with psychotropic drugs is started are very scarce in the young population, which, unlike adults, is generally not influenced by previous psychopharmacological treatments or metabolic pathologies.

In addition, the preliminary results of this study imply the relevance of considering diagnosis and treatment of metabolic comorbidities together with management of psychiatric conditions that require initiation of psychotropic drug therapy. This simultaneous treatment could prevent progression in both difficult-to-treat metabolic pathologies and psychiatric disorders that may respond better to psychotropic drugs [52].

There are several strategies that could be implemented in clinical practice to prevent or reduce the impact of metabolic risk, including knowledge of the psychotropic drugs that increase this risk more than others; adaptation of drug prescriptions to the cardiovascular risk profiles of patients, especially of those already vulnerable; reduction in drug dose; changes in drug type; and monitoring of metabolic risk to provide timely treatment [52,53]. Metformin plus statins or topiramate may be effective in treating psychotropic-drug-induced metabolic side effects [54,55,56]. Modifying the lifestyles of psychiatric patients who take psychotropic drugs (cognitive and behavioral interventions, nutritional counseling and exercise), even preventively, has been shown to improve the severity of depression and psychotic symptoms, as well as to modify and reduce metabolic risk [57,58,59]. Another intervention strategy hypothesizes the addition of anti-inflammatory agents, such as nonsteroidal anti-inflammatory drugs or N-acetylcytein, that could have direct psychiatric-symptom-reducing effects or improve the efficacy of psychotropic medications [60,61,62]. The gut is colonized by trillions of microorganisms, called microbiota, that play vital roles in metabolism, immunity and neurobiology. Penetration of bacteria through the intestinal epithelium could modulate neurotrophins and proteins that are involved in brain development and plasticity, resulting in chronic inflammation, which would further worsen metabolic risk. Recent metagenomic research has indicated that compositional changes in gut microbiota could be contributing factors to development of several neuropsychiatric diseases, such as depression, suggesting that psychobiotics and pharmabiotics could be used to both maintain balanced compositions of gut microbiota and improve the symptoms of these disorders [63,64].

Metagenomics studies, together with recent preclinical research, such as studies that evaluate the role of the mitochondrial tryptophan–kynurenine metabolic system and its connection with clinical manifestations of several neuropsychiatric disorders, are contributing to identification of potential biomarkers for understanding the typical biologies of these disorders, exploration of their pathogeneses, determination of the differences between individuals who suffer from them, treatment choices and support of current diagnostic methods [65,66,67].

Future research should consider the importance of multiplying studies of potential biomarkers that may predispose metabolic risk during treatment with psychotropic drugs. Especially for the developmental age, it is necessary to recruit larger samples of patients; stratify the enrolled subjects on the basis of shared features, including age, genre, stage of development and type of psychiatric condition; and plan longer follow-up in order to identify common biomarkers that could be systematically used in normal clinical practice [68].

The ultimate goal is to arrive at a “precision medicine” that identifies effective therapeutic approaches based on patient phenotypes. In this regard, computational medicine has begun to employ artificial intelligence, machine learning and deep learning to elaborate bioinformatics databases, apply algorithms and analyze big data at the molecular, cellular and organic levels [65].

In consideration of the young population, prescription and use of psychotropic drugs always involve problems of an ethical and legal nature and require extreme caution. Knowledge of the possible effects that psychiatric drugs exert on the central nervous system and on bodies that are undergoing rapid transformation and evolution, especially if prescribed for a long time, continues to be limited. Furthermore, it should be taken into account that these drugs could cause interference in the spontaneous changes characteristic of the physiological process of adolescence and aggravate a state of coenesthetic alarm that is related to the threatening experiences that these bodily changes could cause. Investing resources in this line of research is highly important to be able to personalize therapeutic measures for these vulnerable patients.

## 5. Conclusions

Despite the very small size of our sample, we believe in the importance of disclosing the results of this study because it is generally very difficult to enroll and study drug-naïve adolescent patients who are long-term compliant with psychopharmacotherapy. Early diagnosis and institution of appropriate management for the metabolic consequences associated with psychotropic drug use in young people, such as weight gain and insulin resistance, will increase if monitoring is incorporated into routine health care; in fact, this is important before and during treatment, especially in the presence of pre-existing conditions of vulnerability, such as, in the case of our study, a certain glucose profile in the presence of a diagnosis of depression treated with sertraline. Biomarker study has the potential to tailor therapeutic individual interventions to deliver maximum benefits. Therefore, the next step is to personalize psychotropic-drug therapy while trying to obtain the best response and maximum safety. To achieve this result, there are at least three main challenges:-To organize specific guidelines that identify specific biomarkers for evaluation and according to which time intervals they should be evaluated during psychopharmacological treatment;-To identify patients who have the best chance of treatment success, not only from a psychiatric but also a somatic point of view;-To organize specific prevention and intervention programs for each type of drug- induced dysmetabolic profile.

Although there are several promising data, future longitudinal and experimental studies should be carried out.

## Figures and Tables

**Table 1 biomedicines-11-00048-t001:** Sociodemographic and clinical data of both groups.

		Risperidone (n = 22)	Sertraline (n = 18)
**Age Range (Mean ± SD)**		12–17 (12.7 ± 2.8)	12–17 (14.4 ± 1.5)
Gender	Male N (%)	12 (54.5%)	3 (16.7%)
	Female N (%)	10 (45.5%)	15 (83.3%)
Metabolic Disease Family History	Absent N (%)	17 (77.3%)	14 (77.8%)
	Hypertension N (%)	3 (13.6%)	2 (14.2%)
	Diabetes N (%)	2 (9.1%)	3 (21.4%)
	Metabolic Syndrome N (%)	-	1 (7.1%)
	Obesity N (%)	1 (4.5%)	1 (7.1%)
Main Diagnosis	Schizophrenia N (%)	10 (45.5%)	-
	Behavioral Disorders N (%)	8 (36.4%)	-
	Bipolar Disorders N (%)	4 (18.2%)	-
	Major Depressive D. N (%)	-	5 (27.8%)
	Persistent Depressive D. N (%)	-	6 (33.3%)
	Social Anxiety N (%)	-	4 (22.2%)
	OCD N (%)	-	3 (16.7%)
Dose	T0 (mean ± SD)	1 mg/die	50 mg/die
	T1 (mean ± SD)	1.5 mg/die	75 mg/die
Follow-Up Length		4.5 months	5.7 months

“-“ means that there are no patients with that diagnosis.

**Table 2 biomedicines-11-00048-t002:** Baseline comparisons of the anthropometric and metabolic parameters between the two groups.

	Risperidone				Sertraline		
	N	Mean (SD)	Median (IQR)	N	Mean (SD)	Median (IQR)	*p*-Value
Weight	22	51.650 (14.212)	54.500 (18.750)	18	51.583 (8.860)	51.000 (13.750)	0.986
WC	22	69.636 (17.140)	69.000 (22.250)	18	63.111 (11.119)	63.500 (8.500)	0.172
WHtR	22	0.433 (0.128)	0.445 (0.127)	18	0.391 (0.063)	0.400 (0.047)	0.216
BMI	22	21.785 (4.853)	22.100 (7.180)	18	20.445 (3.896)	19.050 (6.325)	0.350
Glucose	22	81.500 (8.222)	81.500 (6.500)	18	78.500 (8.082)	76.500 (14.250)	0.255
Insulin	22	11.305 (6.704)	11.000 (4.500)	18	12.556 (6.220)	11.950 (6.850)	0.568 ^a^
HOMA-IR	22	1.922 (1.047)	1.860 (1.075)	18	2.392 (1.264)	2.295 (1.355)	0.201 ^a^
Cholesterol Tot	22	148.727 (36.087)	140.500 (37.750)	18	163.778 (23.728)	155.500 (39.000)	0.137
Triglycerides	22	70.364 (28.840)	66.000 (26.250)	18	80.333 (39.718)	71.000 (34.500)	0.362 ^a^
HDL	22	54.591 (11.438)	58.000 (15.250)	17	54.647 (16.332)	52.000 (25.750)	0.800
LDL	22	84.000 (30.944)	78.000 (46.000)	18	92.667 (16.037)	91.500 (16.000)	0.289

*p*-values < 0.05. ^a^ Mann–Whitney used for the comparison.

**Table 3 biomedicines-11-00048-t003:** Paired comparisons between T0 and T1 of both risperidone and SSRIs.

	Risperidone							Sertraline				
	T0			T1				T0		T1		
	N	Mean (SD)	Median (IQR)	Mean (SD)	Median	*p*-Value	N	Mean (SD)	Median (IQR)	Mean (SD)	Median	*p*-Value
Weight	22	51.650 (14.212)	54.500 (18.750)	51.650 (14.212)	56.500 (17.200)	<0.001 ^a^	18	51.583 (8.860)	51.000 (13.750)	54.667 (8.520)	54.500 (11.400)	0.008 ^a^
WC	22	69.636 (17.140)	69.000 (22.250)	74.045 (16.888)	75.000 (16.500)	<0.001 ^a^	18	63.111 (11.119)	63.500 (8.500)	66.583 (10.424)	66.000 (10.000)	0.005 ^a^
WHtR	22	0.433 (0.128)	0.445 (0.127)	0.488 (0.120)	0.485 (0.120)	0.008 ^a^	18	0.391 (0.063)	0.400 (0.047)	0.446 (0.132)	0.425 (0.077)	0.002 ^a^
BMI	22	21.785 (4.853)	22.100 (7.180)	23.908 (4.559)	24.050 (6.160)	<0.001	18	20.445 (3.896)	19.050 (6.325)	21.461 (3.879)	20.900 (5.730)	0.026 ^a^

^a^ Wilcoxon used for the comparison.

**Table 4 biomedicines-11-00048-t004:** Correlations between the differences in each anthropometric variable and the blood biochemical parameters between T1 and T0 for the risperidone group.

		Δ Weight	Δ WC	Δ WtHR	Δ BMI
Glucose	Rho	−0.280	−0.362 ^b^	−0.230 ^b^	−0.296
	2-Tailed p	0.207	0.098	0.303	0.181
Insulin	Rho	−0.341 ^b^	−0.213 ^b^	−0.381 ^b^	−0.167 ^b^
	2-Tailed p	0.121	0.341	0.080	0.457
HOMA-IR	Rho	−0.182	−0.043 ^b^	−0.216 ^b^	−0.158
	2-Tailed p	0.418	0.850	0.335	0.482
Cholesterol Tot	Rho	−0.092	0.014 ^b^	0.232 ^b^	−0.092
	2-Tailed p	0.684	0.950	0.299	0.685
Triglycerides	Rho	−0.078 ^b^	0.007 ^b^	−0.058 ^b^	−0.115 ^b^
	2-Tailed p	0.730	0.975	0.797	0.609
HDL	Rho	−0.267	−0.426 *^,b^	−0.106 ^b^	−0.277
	2-Tailed p	0.229	0.048	0.640	0.212
LDL	Rho	−0.089	−0.013 ^b^	0.170 ^b^	−0.104
	2-Tailed p	0.694	0.954	0.450	0.645

* *p*-values < 0.05 ^b^ Rho di Sperman was used for the correlation.

**Table 5 biomedicines-11-00048-t005:** Correlations between the differences of each anthropometric variable and the blood biochemical parameters between T1 and T0 for the sertraline group.

		Δ Weight	Δ WC	Δ WtHR	Δ BMI
Glucose	Rho	−0.033 ^b^	−0.039	0.020 ^b^	0.147 ^b^
	2-Tailed p	0.896	0.878	0.938	0.561
Insulin	Rho	0.340 ^b^	0.334 ^b^	0.257 ^b^	0.532 *^,b^
	2-Tailed p	0.167	0.175	0.304	0.023
HOMA-IR	Rho	0.292 ^b^	0.296 ^b^	0.171 ^b^	0.482 *^,b^
	2-Tailed p	0.240	0.233	0.496	0.043
Cholesterol Tot	Rho	0.101 ^b^	0.147	0.067 ^b^	−0.066 ^b^
	2-Tailed p	0.689	0.560	0.791	0.794
Triglycerides	Rho	0.149 ^b^	0.157 ^b^	−0.151 ^b^	0.011 ^b^
	2-Tailed p	0.555	0.534	0.550	0.964
HDL	Rho	−0.178 ^b^	−0.199	−0.104 ^b^	−0.223 ^b^
	2-Tailed p	0.480	0.428	0.681	0.374
LDL	Rho	0.343 ^b^	0.286	0.233 ^b^	0.177 ^b^
	2-Tailed p	0.164	0.251	0.351	0.482

* *p*-values < 0.05 ^b^ Rho di Sperman was used for the correlation.

## Data Availability

Not applicable.

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
