# Peer review of "Glucose and Lipid Profiles Predict Anthropometric Changes in Drug-Naïve Adolescents Starting Treatment with Risperidone or Sertraline: A Pilot Study"

_biomedicines, 2022, doi:10.3390/biomedicines11010048_

Round 1
Reviewer 1 Report
The manuscript describes those biochemical factors influencing weight gain in a very specific population group (adolescents) treated with two antidepressant drugs (risperidone or sertraline). The study is well designed, hypothesis and goals are correctly described, and references are updated. I would like to make some recommendations, that should be considered as mere notes (some of them are included in the manuscript):
1. Although it assesses the influence of biochemical factors in weight gain, adolescence is an age in which self-perception of physical beauty can be a determining factor in this population group, or having a negative perception of it as a consequence of treatment.
2. It is not correct to say “statistically significant” but “significant”.
3. Regarding the different diseases diagnosed in both groups, could they have any influence?
4. In tables 3 and 4, SSRI should be changed to sertraline.
5. The study should be carried out in the long-term to achieve definitive conclusions.

Author Response
Dear reviewer,
we thank you for your appreciation and for your comments which certainly contribute to improving the quality of this manuscript. Here below we list the answers to your comments:
Although it assesses the influence of biochemical factors in weight gain, adolescence is an age in which self-perception of physical beauty can be a determining factor in this population group, or having a negative perception of it as a consequence of treatment.
As you rightly suggested, we included in the Conclusion section such consideration about the adolescence age.
It is not correct to say “statistically significant” but “significant”.
As you suggested, we have eliminated the term “statistically”.
Regarding the different diseases diagnosed in both groups, could they have any influence?
We agree with your consideration, therefore we have modified the Limitations section of our manuscript.
In tables 3 and 4, SSRI should be changed to sertraline.
As you suggested, we have changed the term SSRI to sertraline in tables.
The study should be carried out in the long-term to achieve definitive conclusions.
We have modified the Conclusion section according to your clarification.
Kind regards,
Emilia Matera

Reviewer 2 Report
This is an obsarvational study, where I frankly struggle to find what is the message and innovativeness.
1. observational type of the study without control for lifestyle behaviour is not capable to bring anything new over the studies that reported these metabolic consequences for the treatments previously. Whatever the results are, it can become just one of many other other studies on the same topic, with suboptimal methodology to generate a conclusion.
2. the study is far too small and thus underpowered to consider the results as sufficiently reliable. Moreover, short observational period is other complication.
3. Furthermore, the study has been conducted in period, where there still have been substantioal changes in lifestyles of the families due to COVID weaning off, which could likely bias the findings
4. comparison between the two groups is misleading, since there it not an appropriate control
I do also have further comments to data analysis and presentation:
5. If you use NONPARAMETRIC tests (which is correct here!), median + IQR should be presented instead of mean, as this is the correct metrics. The statistics should be improved, as in some parts Students t-test have clearly been used, which is not appropriate, unless you tested for normality of the data in these small populations.
6. Table 1 in description of T0 or T1 groups does not correspond to the text in the aspect that according to the textual description T0 is a baseline, T1 follow up results. Howere there is a dose for T0 in the table, so this needs clarification and allignement.
Author Response
Dear reviewer,
we thank you for your considerations which we consider extremely useful for improving the quality of this and our future studies. Here below we list the responses to your comments. We sincerely hope that you will find them suitable for an eventual publication.
Observational type of the study without control for lifestyle behaviour is not capable to bring anything new over the studies that reported these metabolic consequences for the treatments previously. Whatever the results are, it can become just one of many other other studies on the same topic, with suboptimal methodology to generate a conclusion.
We are aware of the limitations of the study you reported and of the fact that the previous studies that have shown metabolic consequences secondary to psychotropic drug treatments in the developmental age are currently more numerous than in the past. Despite this, the literature data on markers predisposing to anthropometric variations at risk of evolution towards metabolic pathologies, before starting treatment with psychotropic drugs, are still very few in the developmental age population.
The study is far too small and thus underpowered to consider the results as sufficiently reliable. Moreover, short observational period is other complication.
We completely agree with this observation. Unfortunately obtain adequate compliance with drug therapy by both patients and their families it is quite difficult. These reasons justify the limited sample size and the short observation period that we have included in the limitations section of the manuscript, although, as shown by previous studies published by our group and by other authors, that we included in the references list of the manuscript, risperidone seems to induce significant increase in weight and variations in anthropometric and metabolic parameters after a few months of therapy in children and adolescent patients. Anyway we are continuing with the collection of further data, therefore those proposed are to be considered preliminary.
Furthermore, the study has been conducted in period, where there still have been substantial changes in lifestyles of the families due to COVID weaning off, which could likely bias the findings
Considering your right observation, we have changed the limitations section of the manuscript.
Comparison between the two groups is misleading, since there it not an appropriate control
Risperidone and Sertraline groups were compared only at baseline, before starting the drug therapy, to highlight that there were no significant differences between the anthropometric and haematochemical parameters examined. However the lack of a control group was reported in the Limitations section of the manuscript.
If you use NONPARAMETRIC tests (which is correct here!), median + IQR should be presented instead of mean, as this is the correct metrics. The statistics should be improved, as in some parts Students t-test have clearly been used, which is not appropriate, unless you tested for normality of the data in these small populations.
As you correctly pointed out, we added the IQR for medians in tables 2 and 3. We specified in the statistical analysis paragraph that the normality of distributions of all variables were assessed through the Shapiro Wilk test, then we used the non-parametric U-Mann Withney test or Wilcoxon test for the non normal distributed variables, while t-test was used to normal distributed variables.
Table 1 in description of T0 or T1 groups does not correspond to the text in the aspect that according to the textual description T0 is a baseline, T1 follow up results. Howere there is a dose for T0 in the table, so this needs clarification and allignement.
We are sorry for the inaccuracy. For this reason, we changed the “Assessment and Measurements” section of the manuscript. The dose indicated in T0 was the dose prescribed after the first blood examination (starting therapy dose); the dose at T1 was the dosage the patient assumed at the moment of follow up.
Kind regards,
Emilia Matera

Round 2
Reviewer 2 Report
Thank you for the revision. There are some minor improvements of the text, however, my concerns on vastly insuficient methodology and lack of novelty still remain (original major concerns No1-3). I do understand this is something that cannot be improved, but it in my view excludes this paper from publication.
Author Response
Dear reviewer,
we are sorry if you found our changes to the manuscript insufficient.
We thank you for your time.
Best regards,
Emilia Matera